# Heterogeneous Electro-Fenton-Catalyzed Degradation of Rhodamine B by Nano-Calcined Pyrite

**DOI:** 10.3390/ijerph20064883

**Published:** 2023-03-10

**Authors:** Yu Tan, Changsheng Zhao, Qingfeng Chen, Luzhen Li, Xinghua Wang, Beibei Guo, Bowei Zhang, Xiaokai Wang

**Affiliations:** 1Shandong Analysis and Test Center, Qilu University of Technology (Shandong Academy of Sciences), Jinan 250014, China; 2College of Geography and Environment, Shandong Normal University, Jinan 250300, China

**Keywords:** electro-Fenton, magnetic pyrite, nanocatalyst, rhodamine B degradation, boron-doped diamond anode, mineralization

## Abstract

The use of natural pyrite as a catalyst for the treatment of recalcitrant organic wastewater by an electro-Fenton system (pyrite-EF) has recently received extensive attention. To improve the catalytic activity of natural pyrite (Py), magnetic pyrite (MPy), and pyrrhotite (Pyr), they were obtained by heat treatment, and the nanoparticles were obtained by ball milling. They were characterized by X-ray diffraction, X-ray electron spectroscopy, and scanning electron microscopy. The degradation performance of rhodamine B (Rhb) by heterogeneous catalysts was tested under the pyrite-EF system. The effects of optimal pH, catalyst concentration, and current density on mineralization rate and mineralization current efficiency were explored. The results showed that the heat treatment caused the phase transformation of pyrite and increased the relative content of ferrous ions. The catalytic performance was MPy > Py > Pyr, and the Rhb degradation process conformed to pseudo-first-order kinetics. Under the optimum conditions of 1 g L^−1^ MPy, an initial pH of five, and a current density of 30 mA cm^−2^, the degradation rate and TOC removal rate of Rhb wastewater reached 98.25% and 77.06%, respectively. After five cycles of recycling, the chemical activity of MPy was still higher than that of pretreated Py. The main contribution to Rhb degradation in the system was •OH radical, followed by ${\mathrm{SO}}_{4}^{•-}$, and the possible catalytic mechanism of MPy catalyst in the pyrite-EF system was proposed.

## 1. Introduction

Dye wastewater is widely sourced from textile, food, medicine, plastics, and other industries [1]. It is rich in color, high in organic content, highly toxic, and difficult to degrade. The main components are aromatic and heterocyclic compounds, which contain chromogenic and polar groups [2]. Some of these organics can not only affect microbial and plant photosynthesis [3], but also have strong carcinogenicity and mutagenicity for humans [4]. Traditional dye wastewater treatment technologies include adsorption, coagulation, flocculation, and biochemical methods, but there are disadvantages such as secondary pollution and low treatment efficiency [5]. Previous studies showed that advanced oxidation processes (AOPs) were commonly used to degrade such recalcitrant compounds [6], which were completely mineralized into carbon dioxide, water, and inorganic ions by strong oxidizing hydroxyl radicals (•OH), sulfate radicals (${\mathrm{SO}}_{4}^{•-}$), etc. [7,8]. 

Among them, electrochemical advanced oxidation processes (EAOPs) have received extensive attention due to their high processing efficiency, low reagent consumption, and simple operation [9]. At present, the most researched aspects of EAOP are anodic oxidation (AO) and electro-Fenton (EF) [10]. AO is based on the reaction (1) to generate heterogeneous hydroxyl radicals M (•OH) [11,12] on the surface of the anode material (M) through the water oxidation reaction to degrade organics. In addition, anodes with high oxygen evolution overpotential, such as boron-doped diamond (BDD), etc., can also generate oxidants such as persulfate (S_2_O_8_^2−^) through reactions (2) in the case of using SO_4_^2−^ as the electrolyte to participate in the mediated oxidation processes [13,14]. The EF process needs acidic conditions (pH of 2.5~3.5) to prevent iron precipitation [15]. The introduced O_2_ gas undergoes an electroreduction reaction (3) at the cathode, continuously generating H_2_O_2_, which undergoes a Fenton reaction (4) with the Fe (II) added from an external source. Fe (Ⅱ) can be continuously generated at the cathode through the electroreduction reaction (5) of Fe (Ⅲ),catalyzing the oxidation of organic matter [16]. At the right pH, EF overcomes the massive dosing of oxidants and the solid sludge problem caused by the loss of iron ions in the traditional Fenton system [17]. The current main research directions are the synthesis of new electrode materials and the development of heterogeneous Fenton catalysts supported by iron sources. Novel electrodes such as ferrite carbon aerogel cathodes [18], CoFe hydroxide, and iron oxide-coated carbon felt cathodes [19,20] rely on the iron source supported on the electrode to replace the traditional graphite electrode. Novel catalysts such as sepiolite (for adsorbing iron sources) [21], goethite, and lepidocrocite, with nano-iron oxyhydroxide supported on activated carbon [22], utilize cheap natural materials to reduce catalyst costs.
(1)$$M+H_{2}O\;\to \;M\;(•\mathrm{OH})+H^{+}+e^{-}$$
2SO_4_^2−^ → S_2_O_8_^2−^ + 2e^−^(2)
O_2_ + 2H^+^ + 2e^−^ → H_2_O_2_(3)
(4)$${\mathrm{Fe}}^{2+}+H_{2}O_{2}\;\to \;{\mathrm{Fe}}^{3+}+•\mathrm{OH}+{\mathrm{OH}}^{-}$$
Fe^3+^ + e^−^ → Fe^2+^(5)

Pyrite (FeS_2_) is a natural iron-bearing mineral that can participate in AOPs, and is commonly found in the Earth’s crust and the intermediate products generated in water [23]. The current research on pyrite-mediated AOPs is mainly divided into two categories: First, the pyrite-Fenton system not only uses pyrite as an iron source catalyst but also provides necessary acidic conditions for Fenton through chemical reactions (6~8) with water and hydrogen peroxide [24,25,26]. Second, the pyrite-activated persulfate (pyrite-PMS) system based on Formula (9) promotes the generation of ${\mathrm{SO}}_{4}^{•-}$ radicals to oxidize organic pollutants [27,28]. Recently, an electro-Fenton system using pyrite (pyrite-EF) as a catalyst has been used to degrade organic pollutants such as antibiotics [29,30], tyrosol [31], etc. However, pyrite has limitations such as insufficient dispersibility and surface reactivity, slow dissolution, limited efficiency of heterogeneous reactions at the solid-liquid interface, and low catalytic activity [32]. Strategies such as nanopyrite [33,34,35], microwave [36], ultrasound [37], and light-assisted [38] were used to enhance the performance of pyrite-mediated AOPs. It has been reported that pyrrhotite (Fe_1−x_S) has a higher iron content than pyrite, and its specific ferromagnetic properties can promote phase separation [39]. Pyrrhotite can also be used as a Fenton reaction catalyst [40] and a persulfate activator [41,42]. Apart from the above method, heat treatment can also improve the chemical activity of pyrite, reduce the crystal size of the calcined product [43], and increase the specific surface area. However, currently, heat-treated pyrite is commonly used in the field of adsorption of phosphates [44,45] and heavy metals [46]. The use of heat-treated pyrite as a Fenton-like catalyst has not yet been reported.
2FeS_2_ + 7O_2_ + 2H_2_O→ 2Fe^2+^ + 4SO_4_^2−^ + 4H^+^(6)
2FeS_2_ + 15H_2_O_2_ → 2Fe^3+^ + 14H_2_O + 4SO_4_^2−^ + 2H^+^(7)
FeS_2_ + 14Fe^3+^ + 8H_2_O → 15Fe^2+^ + 2SO_4_^2−^ + 16H^+^(8)
(9)$${\mathrm{FeS}}_{2}+2{S_{2}O_{8}}^{2-}\;\to \;2{\mathrm{SO}}_{4}^{•-}+{\mathrm{Fe}}^{2+}+2{\mathrm{SO}}_{4}^{2-}+2S$$

In this paper, differently modified catalysts were obtained by pretreatment and heat treatment of natural pyrite and ball-milled to nano-sized particles as a heterogeneous Fenton catalyst to degrade Rhodamine B (Rhb) using a pyrite-EF system with a BDD anode. Characterization revealed the phase and composition changes of the catalyst before and after heat treatment and reaction. The degradation ability of Rhb by electrocatalysis (EC), traditional EF, and catalysts before and after heat treatment was investigated, and the effects of catalyst concentration, pH, and current density on mineralization rate and mineralization’s current efficiency were explored. The possible catalytic mechanism of the MPy catalyst in the EF-pyrite system was proposed.

## 2. Materials and Methods

### 2.1. Materials

Natural pyrite mineral (millimeter scale) was purchased from Anhui, China. Rhodamine B (Rhb, AR, C_28_H_31_CIN_2_O_3_) and tert-butanol (TBA, AR, C_4_H_10_O) were purchased from Macklin (Shanghai, China). Absolute ethanol (AR, C_2_H_6_O) was purchased from Kermel (Tianjin, China). Phenol (Phenol, AR, ≥99.7%, C_6_H_6_O), ferrous sulfate heptahydrate (AR, FeSO_4_·7H_2_O) and other chemical reagents (analytical grade) were purchased from Sinopharm Chemical Reagent Co., Ltd. (Shanghai, China).

### 2.2. Sample Preparation

Natural pyrite was pickled with 10% HCl for 30 min to remove surface oxide impurities, rinsed with deionized water, and then dried in an electric blast drying oven (101 series, Yongguangming, Beijing, China) at 80 °C for 12 h to obtain pretreated pyrite (Py). Heat treatment was the way of open-atmosphere calcination. An appropriate amount of pretreated pyrite was placed in a crucible for calcination in the muffle furnace (XL-2006, Tianjian, China). The calcination temperature and time were adjusted to 700 °C and 1 h to obtain magnetic pyrite (MPy) and 800 °C and 1 h to obtain pyrrhotite (Pyr). Then, all samples (Py, MPy, and Pyr) were ground in a high-energy planetary ball mill (YXQM-20L, MITR, Changsha, China) at a speed of 200 rpm for 6 h to obtain nano-sized particles. The mass ratio of the grinding balls (3 mm, 6 mm, and 10 mm in diameter) to the samples was 20:1, and ethanol was selected as a grinding aid (4 mL/5 g). Finally, the as-milled powder (nano-samples) was dried and sealed for storage.

### 2.3. Rhb Removal Experiments

An electrolytic cell with a capacity of 250 mL was used. The volume of the electrolyzed Rhb solution (100 mg L^−1^) was 150 mL, and 1.5 mL Na_2_SO_4_ (1 mol L^−1^) was added as electrolyte. The initial pH value was adjusted with 0.25 mol L^−1^ NaOH and H_2_SO_4_ solutions ranging from 3 to 9. Then the heterogeneous nanocatalysts Py, MPy, and Pyr were added with concentrations ranging from 0.5 to 3 g L^−1^, respectively. An adjustable power supply (MS-303D, MAISHENG, Dongguan, China) was used to apply a constant current ranging from 10 to 40 mA cm^−2^. BDD anode (30 × 30 × 0.52 mm) and graphite cathode (30 × 45 × 2 mm) were purchased from Saiao Electrochemical Instrument Co., Ltd. (Saiao, Hangzhou, China), and the electrode plate spacing was 2 cm. The entire electrolysis process was carried out at room temperature. The magnetic stirrer (DLAB, Shanghai, China) was set at 500 r min^−1^ to suspend the catalyst in the solution and reduce agglomeration. The continuous air flow was 50 mL min^−1^. No catalyst was added to the EC experiment. In the EF experiments, FeSO_4_·7H_2_O with the same iron content as pyrite was added as a catalyst. In the catalyst stability experiment, the used catalyst was separated from the solution by applying an external magnetic field, washed with deionized water, and dried in the drying oven to be used as the catalyst for the next batch of experiments. In the free radical recognition experiment, anhydrous ethanol, tert-butanol, and phenol were used as free radical quenchers to determine the reactive oxygen species in the EF-pyrite system.

### 2.4. Instruments and Analytical Procedures

The concentration of Rhb was measured at 554 nm using a UV spectrophotometer (759s, Xipu, Shanghai, China). Total organic carbon was measured by the total organic carbon analyzer (TOC-2000, METASH, Shanghai, China). The surface morphologies of the pyrite samples were analyzed by performing scanning electron microscopy (SEM, Zeiss Sigma 300, Oberkochen, Germany), and the material compositions were analyzed by X-ray diffraction (XRD, Rigaku Ultima IV, Tokyo, Japan). The chemical states of the elements in the pyrite samples were analyzed by X-ray photoelectron spectroscopy (XPS, Thermo Scientific K-Alpha, Shanghai, China). The reaction rate constant *k* (min ^−1^) was calculated according to Equation (10):(10)$$k\cdot t=\ln\left(C_{0}/C_{t}\right)$$

Mineralization current efficiency (MCE) is a parameter that reflects the effective chemical mineralization reaction in the process of an electrochemical reaction. In the experiment, the removal of TOC per unit time was used as the effective reaction in the process, and the calculation Formula (11) was as follows:(11)$$MCE\left(\%\right)=\frac{nFV\Delta \left(TOC\right)}{4.32\times 10^{7}\;mIt}\times 100$$
where *F* is the Faraday constant (96,487 C/mol), *V* is the solution volume (L), Δ(*TOC*) is the reduction in TOC during the degradation process (mg L^−1^), and 4.32 × 10^7^ is the homogeneity factor (=3600 s/h × 12,000 mg), *m* is the number of carbon atoms in the Rhb molecule, *I* is the applied current (A), and *t* is the electrolysis time (h). The number of electrons *n* consumed during the mineralization of Rhb is taken as 146, according to its conversion to CO_2_, chloride, and nitrate as the main ions, as described below in reaction (12):C_28_H_31_ClN_2_O_3_ + 59H_2_O → 28CO_2_ + 149H^+^ + Cl^−^ + 2NO_3_^−^ + 146e^−^
(12)

## 3. Results and Discussion

### 3.1. Catalysts’ Characterization

Past studies have shown that pyrite can be thermally treated to change the mineral structure and phase [47] and increase the surface oxidation rate to increase the chemical activity [48]. The XRD patterns of the pretreated Py, the thermally treated MPy, and the Pyr and MPy after the first reaction are shown in Figure 1a. The XRD patterns of samples Py and Pyr indicated pyrite phases (pyrite: FeS_2_ JCPDS: 42-1340) and pyrrhotite phases (Pyrrhotite-4H: Fe_1−x_S JCPDS: 22-1120), respectively. When the calcination temperature reached 700 °C, the pyrrhotite crystal plane appeared in the XRD spectrum, indicating that the pyrite began to undergo phase transformation, but the pyrite crystal plane still existed, indicating that MPy was a complex containing Py and Pyr. When the temperature reached 800 °C, all the diffraction peaks were pyrrhotite crystal planes, indicating that the phase had been completely transformed into pyrrhotite. A previous study showed that the heat treatment temperature increased the magnetic susceptibility to a maximum at 700 °C. At 800 °C, the magnetic susceptibility decreased [49]. After the reaction, most of the pyrrhotite crystal planes in MPy disappeared, and the 208-crystal plane of MPy was further desulfurized and transformed into monoclinic pyrrhotite and iron with 2θ of 43° and 44°. MPy still maintained the original magnetism. On the one hand, it might be that the pyrrhotite crystal face of MPy was consumed in the reaction, and on the other hand, it might be that the reverse phase transition of pyrrhotite to pyrite occurred in the electrochemical process.

X-ray photoelectron spectroscopy characterized the chemical composition and electronic state of surface elements in all samples. The XPS full-spectrum Figure 1b showed that Fe (2p) and S (2p) signal peaks were observed at about 711 and 163 eV. The high-resolution spectra of Fe (2p) and S (2p) were shown in Figure 1c–f. The peaks at binding energies 707 and 710 eV corresponded to Fe (II), and 713 eV corresponded to Fe (III) [50]. The peaks at binding energies of 162 and 163 eV corresponded to S_2_^2−^ and S_n_^2−^, representing pyrite and polysulfide, respectively. The two peaks at 168 and 169 eV corresponded to SO_4_^2−^ [51]. After high-temperature heat treatment at 700 and 800 ℃, the binding energy peaks at 710 and 713 eV of pyrite were relatively decreased, and the peak at 707 eV was increased, indicating that part of the Fe (Ⅲ) was reduced to Fe (Ⅱ). At the same time, the peak intensities of the binding energies 168 and 169 eV decreased, and the peaks at 162 and 163 eV increased, indicating that SO_4_^2−^ in the form of alkali metal and alkaline earth metal sulfates in natural pyrite was reduced to S_2_^2−^ and S_n_^2−^ [52].

The relative content of Fe (II) at the 707 and 710 eV peaks after the MPy reaction increased by 3.20% and 8.38%, respectively, and the relative content of Fe (Ⅲ) at the 713 eV peak decreased by 12.39%. This might be because the electrochemical process provided exogenous electrons [53], which promoted the natural oxidation process inside pyrite [54]. At the same time, a significant decrease in the peak intensities at 162 and 163 eV and an increase in the peak intensities at 168 and 169 eV were observed in the S (2p) spectrum. This was consistent with the previous study [55]: the change of S (2p) promoted the regeneration of Fe (Ⅱ) by S as an electron source on the MPy surface, but under acidic conditions, S atoms in the MPy more easily reacted with O atoms in water to form sulfates, which existed in the MPy or were dissolved in solution [56,57].

The surface morphologies of different catalysts were observed by scanning electron microscopy. Figure 2a showed the highly uniform distribution of pyrite particles, which confirmed the successful conversion of pyrite microparticles into nanoparticles using a high-energy planetary ball milling process. Figure 2c showed that MPy agglomerated after heat treatment at 700 °C, the inner lump crystallized to pyrrhotite, and the surface was covered with powdery pyrite. According to the XRD characterization result, it showed the phase transition of pyrite from the inside to the outside, and the MPy was a composite intermediate product of pyrite and pyrrhotite. It can be seen from Figure 2b that the pyrite has been transformed into pyrrhotite, showing a bulkier crystal structure. Figure 2d–f showed the morphology of the MPy after the reaction. Figure 2d showed that the MPy was dispersed in pyrite form and that the surface of the pyrite was eroded. In addition, a small amount of monoclinic pyrite was produced in MPy after the reaction in Figure 2e. And in Figure 2f, the MPy changed back to pyrite from the outside to the inside during the electrochemical process. This was consistent with the results of the previous XRD characterization.

### 3.2. Effects of Operating Parameters on Rhb Degradation

Figure 3a showed the degradation effect on Rhb of EC, EF, and different catalysts in the pyrite-EF system under the same conditions (catalyst concentration of 1 g L^−1^, an initial pH of 5, and a current density of 30 mA cm^−2^). Previous characterization indicated that MPy had a high ferrous ion content and magnetic susceptibility. Under optimal conditions, the degradation rate of 100 mg L^−1^ Rhb solution catalyzed by MPy was 97.85% within 40 min, which was higher than Py and Pyr. In fact, the catalytic performance advantage of MPy was even greater at shorter treatment times. Figure 3b showed that the catalytic rate constants of MPy were the highest (*k* = 0.12 min ^−1^, *R*^2^ = 0.9771), which were 1.35 times and 1.66 times those of Py and Pyr, respectively. Studies had shown that Fe (II) with binding energies of 707 and 710 eV played an important role in the pyrite-Fenton system [58], so MPy has the best catalytic performance. However, Fe (II) here was more likely to be preferentially dissolved and consumed, resulting in a decrease in the total iron content in the catalyst. However, the relative content of Fe (II) in pyrite was replenished during the electrochemical process. This was the reason why the efficiency of Rhb degradation by Pyr was first higher than that of Py and then lower than that of Py. Moreover, the degradation rate constants of both pretreated and heat-treated pyrite were higher than those of EC and EF. The traditional EF had a faster degradation rate of Rhb in the first 10 min, which was because the traditional EF added ferrous ammonium sulfate with the same iron quality as pyrite. However, in the subsequent degradation process, the degradation rates of Rhb by the pretreated and heat-treated nanopyrites were higher than those of traditional Fenton catalysts. Due to the coexistence of Fe (II) and Fe (III) in pyrite, the iron cycle was promoted to accelerate the Fenton reaction rate. 

The current density had a significant effect on the degradation of Rhb in the EF-pyrite system. The effect of current density (10~40 mA cm^−2^) on the degradation of Rhb under optimal conditions (1 g L^−1^ MPy and an initial pH of 5) is shown in Figure 4a. When the current density was increased from 10 to 30 mA cm^−2^, the final degradation rate of Rhb increased by 30.08%. The kinetic constant was increased from 0.0244 to 0.12 min^−1^, an increase of 5.31 times. This was because the increase in current promoted the formation of BDD (•OH) and S_2_O_8_^2−^ in the anodic reactions (1 and 2). At the same time, the production of H_2_O_2_ in reaction (3) on the cathode and the regeneration of Fe (II) in reaction (5) were accelerated, promoting the generation of •OH radicals in the Fenton reaction (4) and ${\mathrm{SO}}_{4}^{•-}$ radicals in reaction (9), which greatly improved the degradation rate of Rhb. Figure 4b showed that TOC gradually decayed with electrolysis time and increased with increasing current. Under the current density of 10, 20, 30, and 40 mA cm^−2^, the TOC in the Rhb solution was removed by 54.50%, 68.10%, 77.06%, and 84.78%, respectively, within 4 h. Although the removal rate of TOC increased with the increase in current intensity, the MCE decreased by 44.07%, 16.25%, 10.38%, and 11.87%, respectively, after 4 h of electrolysis. When the current increased, the decay of the current efficiency and the tendency to decrease the reaction rate constant were unavoidable [59]. On the one hand, it was due to the self-consumption of BDD (•OH) produced in excess by the anodic reaction (14) [60]. On the other hand, the oxidant loss was caused by electron-hole recombination. At the same time, due to the high current intensity, the hydrogen evolution and oxygen evolution reactions [61] and heat generation at the cathode and anode reduced the current efficiency, resulting in an increase in energy consumption costs.

The effect of an initial pH of 3 –9 on the degradation of Rhb in the EF-pyrite system (current density: 30 mA cm^−2^, 1 g L^−1^ MPy) was investigated. Figure 5a showed that the degradation effect of Rhb generally decreased with an increase in the initial pH. The optimal pH of the traditional Fenton reaction was about 3. Since pyrite provided acidic conditions, the removal rate of Rhb was highest at an initial pH of 5. The kinetic constant of Rhb degradation at pH = 5 was significantly higher than that at pH = 7 (*k* = 0.0488 min^−1^) and pH = 9 (*k* = 0.0374 min^−1^) and slightly lower than that at pH = 3 (*k* = 0.1344 min^−1^) in the first 25 min. At higher pH, H_2_O_2_ was preferentially decomposed to O_2_ rather than •OH radicals [62], and the production of iron sludge was not conducive to iron ion cycling. Due to the surface-catalyzed reaction, the degradation rate of Rhb at an initial pH of 3 was higher than that at a pH of 5 during the first 15 min of the reaction [63], but the •OH radicals and Fe (II) reacted (14) with excess H^+^ and were consumed [64]. Moreover, it was of great significance to reduce the adjustment of initial pH in the actual application process.
(13)$$2\mathrm{BDD}\;(•\mathrm{OH})\;\to \;2M+O_{2}+2H^{+}+2e^{-}$$
(14)$${\mathrm{Fe}}^{2+}+•\mathrm{OH}+H^{+}\;\to \;{\mathrm{Fe}}^{3+}+H_{2}O$$
(15)$${\mathrm{Fe}}^{2+}+•\mathrm{OH}\;\to \;{\mathrm{Fe}}^{3+}+{OH}^{-}$$
(16)$$2•\mathrm{OH}\;\to \;H_{2}O_{2}$$
(17)$$H_{2}O_{2}+•\mathrm{OH}\;\to \;HO_{2}+H_{2}O$$

The catalyst concentration was also an important parameter affecting the Fenton reaction (4), as the heterogeneous Fe (II) concentrations in the solution affected the rate of •OH radical generation [65]. Using a BDD/graphite electrolytic cell at a constant current density of 30 mA cm^−2^ and an initial pH of 5, the effect of MPy concentrations ranging from 0.5 to 3 g L^−1^ on the degradation rate of Rhb was investigated. The results are shown in Figure 5b. When the MPy concentration was increased from 0.5 to 1 g L^−1^, the final degradation rate of Rhb only increased by 4.06% within 40 min. This is because Fe (II) was recycled in the system and fully utilized in long-term treatment. The kinetic constant of the reaction was increased by 1.55 times, which was because increasing the concentration of Fe (II) in the solution increased the generation rate of •OH radicals and accelerated the degradation of Rhb. However, as the catalyst concentration continued to increase, the degradation rate of Rhb decreased, which was due to the Fe (II) reaction (14 and 15) in excessive catalyst and the side reactions (16) and (17) caused by excess •OH radicals that wasted a lot of reactive oxygen species. 

### 3.3. Catalyst Stability

The MPy catalyst was reused in the pyrite-EF system, as shown in Figure 6a. After five runs, the Rhb removal rate decreased from 98.25% to 96.17%. This was attributed to supplying the electrons required for iron cycling during the electrochemical process. Therefore, the removal rate of Rhb did not decrease significantly after the catalyst was used many times. However, the reaction rate constant of the catalyst decreased gradually with the number of uses. According to energy dispersive spectrometer (EDS) relative mass results before and after the catalyst’s first reaction, iron decreased from 61.98% to 61.15%, sulfur decreased from 28.19% to 19.44%, and oxygen increased from 5.76% to 15.16%. This was due to free radical oxidation and acid dissolution on the catalyst surface, resulting in the loss of iron and sulfur and slowing down the regeneration of iron and the generation rate of sulfate radicals. MPy was structurally unstable during EF and phase changed back to pyrite after the reaction, but the magnetism was preserved. Although the loss of sulfur in the catalyst was relatively high during the electrochemical process, the content of iron did not decrease significantly, so that its catalytic performance was still higher than that of the pretreated Py. The results showed that the catalyst exhibited excellent magnetic recovery performance and stable chemical activity in water treatment.

### 3.4. Free Radical Recognition

To determine the active free radicals in the pyrite-EF system with a BDD anode, the effects of three free radical quenchers (i.e., ethanol, TBA, and phenol) on the degradation of Rhb were investigated. Ethanol and tert-butanol can be used as scavengers for •OH radicals and ${\mathrm{SO}}_{4}^{•-}$ radicals [66], but their reaction rates for •OH radicals ($k_{\;\mathrm{ethanol}/•\mathrm{OH}}$ = (1.2~2.8) × 10^9^ M^−1^s^−1^; $k_{\;\mathrm{TBA}/•\mathrm{OH}}$ = (3.8~7.6) × 10^8^ M^−1^s^−1^) were higher than those for ${\mathrm{SO}}_{4}^{•-}$ radicals ($k_{\;\mathrm{ethanol}/{\mathrm{SO}}_{4}^{•-}}$ = (1.6~7.7) × 10^7^ M^−1^s^−1^; $k{\;}_{\mathrm{TBA}/{\mathrm{SO}}_{4}^{•-}}\;$ = (4.0~9.1) × 10^5^ M^−1^s^−1^). The reaction rate constant of phenol to ${\mathrm{SO}}_{4}^{•-}$ radicals ($k{\;}_{\mathrm{Phenol}/{\mathrm{SO}}_{4}^{•-}}\;$ = 8.8 × 10^9^ M^−1^s^−1^) was higher than that of •OH radicals ($k_{\mathrm{phenol}/•\mathrm{OH}}$ = 6.6 × 10^9^ M^−1^s^−1^) [67]. Figure 6b showed the effect of radical scavengers on the degradation of Rhb by pyrite-EF. When ethanol, tert-butanol, and phenol were added to the system, the degradation of Rhb was significantly inhibited compared with no free radical scavenger. Among them, ethanol and tert-butanol significantly inhibited the degradation of Rhb by 46.00% and 40.35%, respectively, indicating that •OH radicals were generated in the system. At the same time, because $k_{\;\mathrm{ethanol}/•\mathrm{OH}}$ was about 3.75 times that of $k_{\;\mathrm{TBA}/•\mathrm{OH}}$, it showed that the degradation of Rhb could be further inhibited by continuing to increase the concentration of ethanol or tert-butanol. However, further increasing the quencher concentration did not significantly improve the inhibition ability of Rhb degradation, which might be due to the inability of the radical quencher to suppress the heterogeneous •OH radical activity inside the anode. The addition of phenol only inhibited Rhb by 11.75%, indicating that ${\mathrm{SO}}_{4}^{•-}$ radical existed in the system and participated in the degradation of Rhb. The results showed that •OH radicals and ${\mathrm{SO}}_{4}^{•-}$ radicals were generated in the pyrite-EF system with BDD/graphite electrode, of which •OH radical was the main active oxygen species.

### 3.5. Catalytic Oxidation Mechanism of the Pyrite-EF System

In a previous study [62], the nanocatalyst had a certain adsorption capacity, which could adsorb Rhb on the surface of the catalyst for heterogeneous catalytic mineralization. Although the iron in the catalyst was released into the solution due to the dissolution of the acid, the loss was minimal. Therefore, the degradation process of Rhb was dominated by heterogeneity. From the free radical recognition experiment in the pyrite-EF system with a BDD anode, it could be known that the degradation of Rhb mainly depended on •OH radical, followed by ${\mathrm{SO}}_{4}^{•-}$ radical. Based on this, a possible MPy electro-Fenton catalytic mechanism was proposed, as shown in Figure 7. •OH radical was mainly produced by a heterogeneous MPy-catalyzed Fenton reaction. H_2_O_2_ was generated by the reaction of exogenous oxygen and water under the action of the current provided by the cathode. Under the action of an anode heterogeneous BDD (•OH) radical, SO_4_^2−^ directly generated a ${\mathrm{SO}}_{4}^{•-}$ radical on the one hand, and on the other hand, it first converted to S_2_O_8_^2−^ and then generated a ${\mathrm{SO}}_{4}^{•-}$ radical under the catalysis of Fe (II) in a heterogeneous catalyst. The electrons required for the reduction of Fe (Ⅲ) to Fe (II) after the reaction of a heterogeneous catalyst were provided by the oxidation of sulfur inside the catalyst and free electrons produced by the cathode. Moreover, in the presence of both S_2_O_8_^2−^ and H_2_O_2_, more reactive oxygen species can be generated in the system [68] to participate in the degradation of Rhb.

## 4. Conclusions

Heat treatment has been shown to enhance the chemical activity of pyrite. Heat treatment can make the pyrite phase change into pyrrhotite gradually and increase the relative content and magnetic susceptibility of ferrous iron in the pyrite. Magnetism can improve the adsorption efficiency, and a high ferrous content can increase the Fenton reaction rate. The nano-scale particles obtained by ball milling further improved the problem of insufficient dispersibility. In the pyrite-EF system with a BDD anode, the degradation performance of the catalyst for Rhb was tested as MPy > Py > Pyr. Under the optimized operating conditions, 97.85% of Rhb was degraded within 40 min, but the mineralization rate was much slower due to the formation of stubborn by-products, and 77.06% of TOC was removed within 4 h. Compared with ordinary pyrite, MPy obtained by heat treatment showed higher catalytic performance. After five cycles of use, there was no significant loss in the degradability of MPy. After the reaction, the loss of sulfur in the MPy was relatively large, and the iron content did not decrease significantly. The free radical recognition experiment confirmed that •OH radical was the main reactive oxygen species while ${\mathrm{SO}}_{4}^{•-}$ radical played a lesser role, so the loss of sulfur had little effect on the performance of the catalyst. MPy-catalyzed pyrite-EF technology provided a new solution for recalcitrant organic wastewater treatment.

## Figures and Tables

**Figure 1 ijerph-20-04883-f001:**
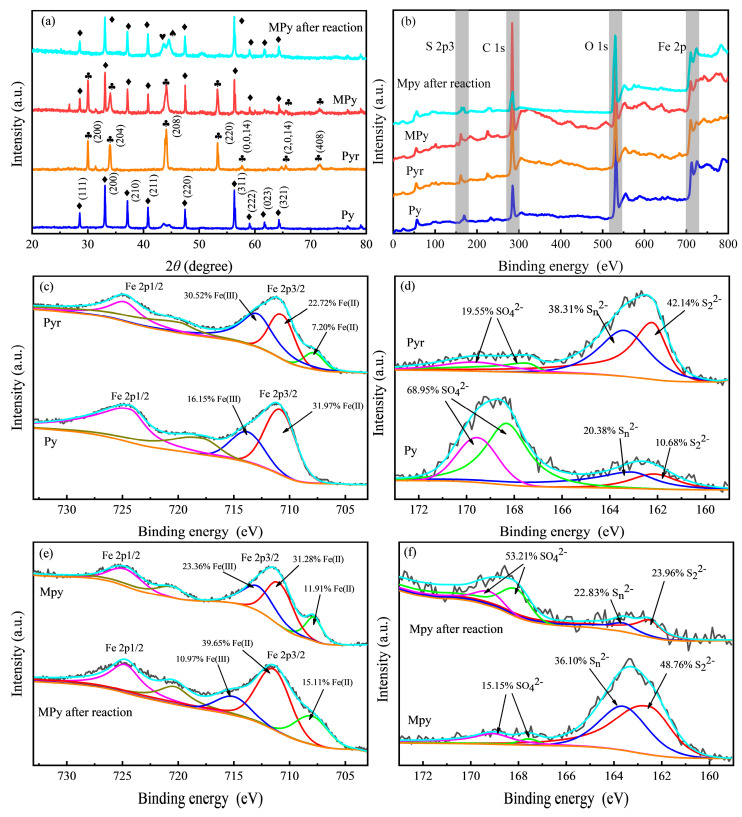
XRD patterns (**a**), XPS full spectra (**b**), and Fe (2p) (**c**,**e**) and S (2p) (**d**,**f**) fine spectra of Py, Pyr, MPy, and MPy after the first reaction. (XRD patterns: ♦ pyrite, ♣ pyrrhotite − 4H, ♥ pyrrhotite − 11T, and ♠ iron).

**Figure 2 ijerph-20-04883-f002:**
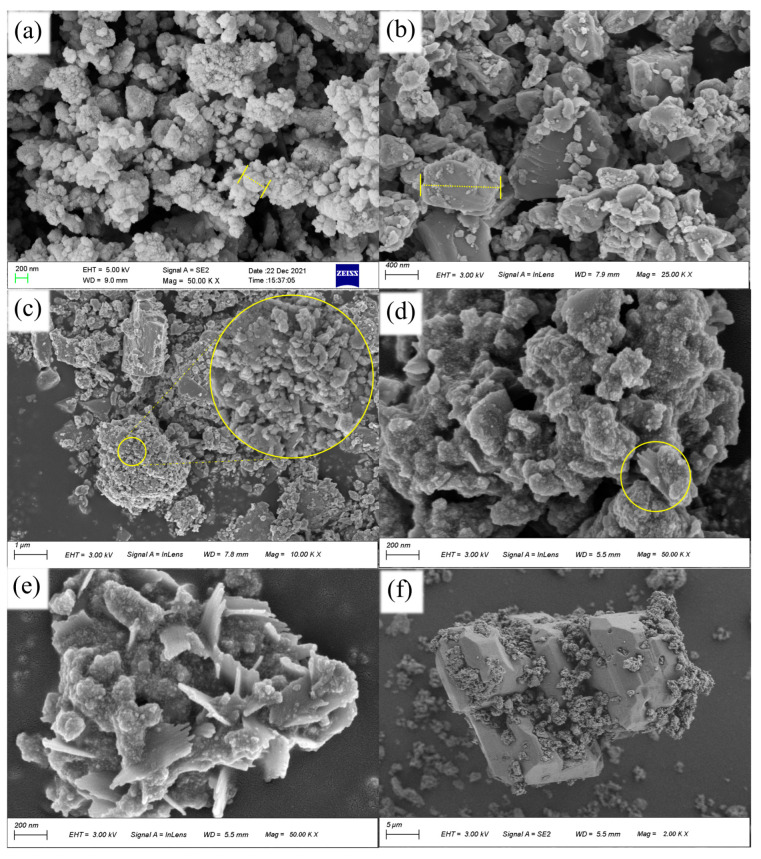
SEM images of Py (**a**), MPy (**b**), Pyr (**c**), and MPy after the first reaction (**d**–**f**) (The scales in a and b are the particle size of the catalyst; The big yellow circle in c is the enlarged picture of the small circle; The circled part in d is pyrrhotite – 11T).

**Figure 3 ijerph-20-04883-f003:**
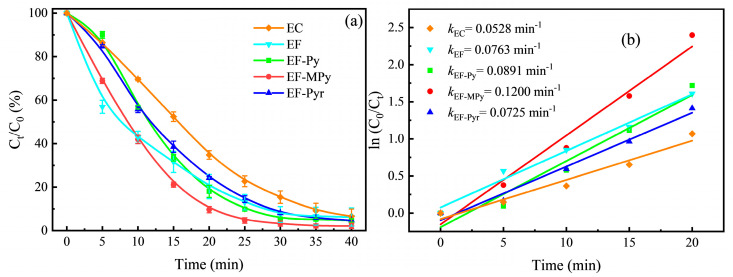
(**a**) The degradation of Rhb with time in electrocatalysis, traditional electro − Fenton, and different catalysts in the pyrite − EF system; (**b**) The first − order kinetic analysis.

**Figure 4 ijerph-20-04883-f004:**
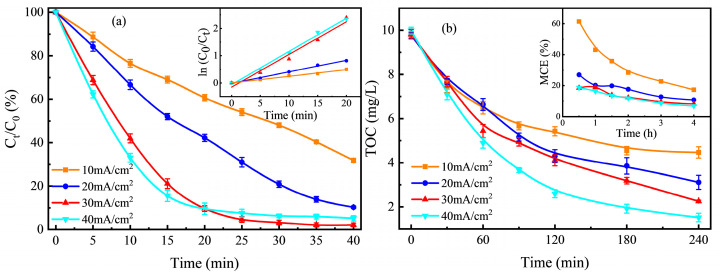
(**a**) Effect of current density on Rhb degradation in the MPy-catalyzed pyrite-EF system and first-order kinetics analysis in the inset panel; (**b**) TOC removal in Rhb solution and MCE (%) value as a function of time in the inset panel.

**Figure 5 ijerph-20-04883-f005:**
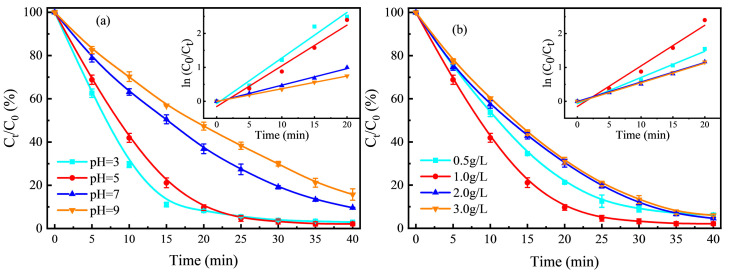
Effects of pH (**a**) and catalyst dosage (**b**) on Rhb degradation in the MPy-catalyzed pyrite-EF system; first-order kinetic analysis of Rhb degradation in the inset panel.

**Figure 6 ijerph-20-04883-f006:**
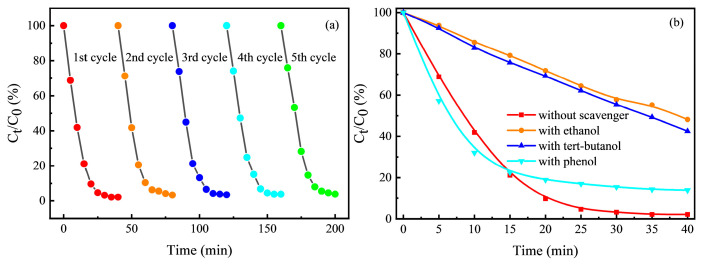
(**a**) Five-cycle recycling performance of MPy in the pyrite-EF system. (**b**) The effect of free radical scavengers on the degradation of Rhb.

**Figure 7 ijerph-20-04883-f007:**
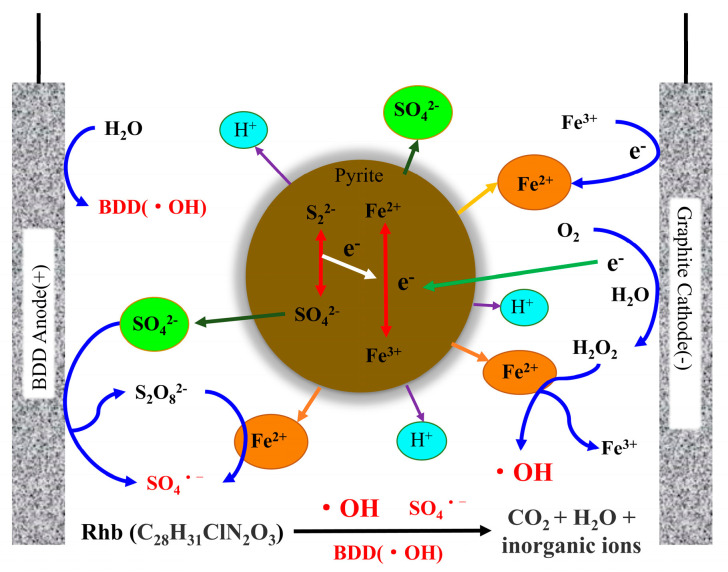
Degradation mechanism in the MPy − catalyzed pyrite − EF system.

## Data Availability

All data analyzed during this study were included in this published article.

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
