# Peer review of "Heterogeneous Electro-Fenton-Catalyzed Degradation of Rhodamine B by Nano-Calcined Pyrite"

_ijerph, 2023, doi:10.3390/ijerph20064883_

Round 1

Reviewer 1 Report

The manuscript presents an adequate development and characterization of both the materials used as catalysts, as well as the reaction system that followed the dye RhB and its mineralization during the degradation process.

I have some questions to solve to complement the information of the manuscript:

1. The composition of the MPy material was Py and Pyr, it was reported in the manuscript  in the X-ray diffraction section (line 175). What phase of iron is for each of them? Is it Fe2+ or Fe3+?

2. Was the iron in the solution measured during the experiments for each catalyst?

3. Figure 1(a) shows the diffraction pattern of the MPy catalyst after the reaction. Was this catalyst used only once?

Was the MPy catalyst characterized after the 5 cycles?

Author Response

Dear editors and reviewers,

We appreciate all the time and efforts you put in the handling of our manuscript entitled “Heterogeneous electro-Fenton-catalyzed degradation of Rhodamine B by nano-calcined pyrite” (Manuscript ID: ijerph-2260314). We also thank the reviewers for your time and dedicated expertise to our manuscript, which greatly improved the quality of the manuscript. We have studied comments carefully and have made corrections which we hope to meet with approval. We use comments in articles to track changes made according to reviewer comments. The responses to the reviewer's comments are as the following “Responses to the reviewer's comments”. We are looking forward to hearing from you soon.

Sincerely,

Changsheng Zhao

Shandong Analysis and Test Center

Qilu University of Technology (Shandong Academy of Sciences)

Keyuan Road 19#

Jinan, Shandong, 250014, China

Tel: +18678817810

Fax: +86-531-82964889

Reviewer 2 Report

1. The reaction (12) includes 146 electrons. Is it in agreement with the reaction of Rhb oxidation in Figure 7? What is oxidant in this reaction?

2.  lines 196-197: ...indicating that SO42- in the form of alkali metal and alkaline earth metal sulfates in natural pyrite was oxidized to S22- and Sn2-

May be "reduced" instead "oxidazed", because the transformation of SO42- ito S22- is reduction process.

3. The surface area influents on both heterogeneous catalytic processes and the rate of leaching of Fe ion into solution. Was the Sbet measured for samples?

4. Please add in Fig 1a the discription of phase signs.

5. Kinetic curves in Fig 3 and 4 was measured with the time interval of 5 min. It can be seen that the kinetic of reaction is limited by diffusion and ather processes after 20 min of reaction. The calculation of konstant of reaction is more correct in first 20 min of reactions.

6. Fig. 6b. Please correct the "ethanol" instead "ethaol".

7. Is the process of Rhd degradation homogeneous or heterogeneous? What is the role of heterogeneous catalyst? Is only Fe2+ ions in solution catalyze the reaction? Is Rhd adsorbed on the catalyst surface? Please, expand the discussion.

Author Response

(The authors gave the same response as above.)
